# Motilin, a Novel Orexigenic Factor, Involved in Feeding Regulation in Yangtze Sturgeon (*Acipenser dabryanus*)

**DOI:** 10.3390/biom14040433

**Published:** 2024-04-02

**Authors:** Ni Tang, Ya Li, Yingzi Li, Shaoqi Xu, Mei Wang, Bin Wang, Yanling Liu, Shupeng Zhang, Hongwei Wu, Xin Zhang, Bo Zhou, Zhiqiong Li

**Affiliations:** 1Department of Aquaculture, College of Animal Science and Technology, Sichuan Agricultural University, Chengdu 611130, China; sautangni1992@163.com (N.T.); liya7911@163.com (Y.L.); 13350051633@163.com (Y.L.); xushaoqi66@163.com (S.X.); 18298352359@163.com (M.W.); lzgo_9221031@163.com (Y.L.); sdzhangshupeng@163.com (S.Z.); whw409@163.com (H.W.); zhangxinscny@163.com (X.Z.); 2Fisheries Research Institute, Sichuan Academy of Agricultural Sciences, Chengdu 610066, China; wangbinhsimple@163.com

**Keywords:** weaning, Yangtze sturgeon, feeding regulation, motilin, food intake, neuroendocrine

## Abstract

Motilin is a gastrointestinal hormone that is mainly produced in the duodenum of mammals, and it is responsible for regulating appetite. However, the role and expression of motilin are poorly understood during starvation and the weaning stage, which is of great importance in the seeding cultivation of fish. In this study, the sequences of Yangtze sturgeon (*Acipenser dabryanus Motilin* (*AdMotilin*)) motilin receptor (*AdMotilinR*) were cloned and characterized. The results of tissue expression showed that by contrast with mammals, *AdMotilin* mRNA was richly expressed in the brain, whereas *AdMotilinR* was highly expressed in the stomach, duodenum, and brain. Weaning from a natural diet of *T. Limnodrilus* to commercial feed significantly promoted the expression of *AdMotilin* in the brain during the period from day 1 to day 10, and after re-feeding with *T. Limnodrilus* the change in expression of *AdMotilin* was partially reversed. Similarly, it was revealed that fasting increased the expression of *AdMotilin* in the brain (3 h, 6 h) and duodenum (3 h), and the expression of *AdMotilinR* in the brain (1 h) in a time-dependent manner. Furthermore, it was observed that peripheral injection of motilin-NH_2_ increased food intake and the filling index of the digestive tract in the Yangtze sturgeon, which was accompanied by the changes of *AdMotilinR* and appetite factors expression in the brain (*POMC*, *CART*, *AGRP*, *NPY* and *CCK*) and stomach (*CCK*). These results indicate that motilin acts as an indicator of nutritional status, and also serves as a novel orexigenic factor that stimulates food intake in *Acipenser dabryanus*. This study lays a strong foundation for the application of *motilin* as a biomarker in the estimation of hunger in juvenile *Acipenser dabryanu* during the weaning phase, and enhances the understanding of the role of motilin as a novel regulator of feeding in fish.

## 1. Introduction

Feeding is crucial for the survival, growth and development of human and animals. During early life, weaning is one of the most important events that involves switching from one type of feed to another feed to meet the nutritional requirement for the growth of mammals and fish [1,2]. In aquaculture, juvenile fish always face a weaning period which involves switching from the consumption of live prey to the consumption of artificial feeds. Usually, fish that fail to be weaned exhibit a decrease in food intake, poor growth, and are susceptible to diseases [3]. The mechanisms underlying the decrease of food intake induced by weaned diets are poorly understood in fish production. Suppressed intake of food may result from the disruption of feeding habits and the change in the nutrient content of the diet. Studies show that nutrient regulation of certain endocrine factors can modulate the feeding and growth of fish [4]. Few studies have explored the feeding endocrine responses of fish during the weaning period. The lack of understanding of the expression of appetite factors induced by weaning limits the capability to develop specific strategies to maintain the performance of fish post-weaning.

Food intake is mainly regulated by peptides and hormones produced by the brain and other peripheral tissues through the neural and endocrine pathways [5]. Studies have reported several factors such as proopiomelanocortin (POMC) [6], neuropeptide Y (NPY) [7], agouti-related protein (AgRP) [8] and cocaine- and amphetamine-regulated transcript (CART) [9] in the brain, leptin in the liver [10], and amylin in the pancreas, which are related to appetite and also regulate food intake in animals. In addition, certain appetite regulation factors that are secreted from the gastrointestinal tract promote (ghrelin) or inhibit (PYY, CCK, GLP-1) food intake [11,12,13,14]. In mammals, it has been reported that AgRP in the brain of swine was not affected; however, MC4R was increased one day post-weaning [15]. Although weaning inhibits food intake, the related changes in appetite factors remain to be determined.

Motilin (MLN) as a polypeptide containing 22 amino acids discovered in porcine duodenal mucosa has been reported to have an orexigenic role in mammals [16]. To date, *motilin* has been identified in several mammals (i.e., humans [16], monkey [17], house musk shrew [18]), birds (i.e., chicken [19], Japanese quail [20]), and amphibians (i.e., Japanese fire belly newts [21]). In mammals, a growing number of biological functions of motilin have been reported such as regulating gastrointestinal motility [22], gastric acid and pepsin secretion [23], left gastric artery diastole [24], circadian rhythm [25], and proliferation and differentiation of adipocyte precursor cells [26]. Motilin primarily binds to G protein-coupled receptor 38 (GPR38), namely the motilin receptor, which contains seven trans-membranes and exerts biological functions [14]. Motilin was demonstrated to exert an orexigenic effect in mammals. In humans, intravenous infusion of motilin significantly promoted food intake [27]. In rats, peripheral administration of motilin significantly increased food intake at 15 min, 1 h and 2 h after injection [28]. In chicks, intracerebroventricular administration of motilin from different animals like porcine, canine or chicken failed to regulate feeding behaviour [29,30]. The information about *motilin* and *motilin receptor* in fish is limited. It has been identified in zebrafish [31,32], ballan wrasse [33] and spotted sea bass [34]. In spotted sea bass, *motilin* and *motilin receptor* expressions were sensitive to fasting [34]. Whether motilin can act as a nutritional indicator in the fish needs to be further explored.

Yangtze sturgeon (*Acipenser dabryanus*) is an endemic freshwater species inhabiting the Yangtze River in China. The wild *Acipenser dabryanus* is difficult to locate and the natural population has declined due to the alteration of habitat, environmental pollution and overfishing [35]. *Acipenser dabryanus* was listed in the International Union for Conservation of Nature and Natural Resources (IUCN) Red List [36,37]. The large scare artificial breeding of wild mature *Acipenser dabryanus* did not succeed until 2003, and then the second generation and third generation were only achieved in 2007 and 2018, respectively. The cultured juvenile and mature *Acipenser dabryanus* released into the wild helped restore the population [38]. In the seeding cultivation of Yangtze sturgeon, especially, the artificial weaning stage, with inappropriate nutrient supply can cause a decrease in food intake and slow growth of juveniles, which limits the restoration of the wild Yangtze sturgeon population. Therefore, it is necessary to explore the molecular mechanism of feeding regulation during different nutritional status especially linked to population recovery. Although the orexigenic effect of motilin in mammals has been studied, whether Yangtze sturgeon motilin functions as an orexigenic indicator involved in feeding regulation needs to be established. Therefore, this study cloned the *motilin* and *motilin receptor* gene in *Acipenser dabryanu*, and evaluated its expression in response to weaning and food deficiency, and explored the role of the motilin peptide on the regulation of feeding. This study will lay a theoretical foundation for understanding the role of motilin in the regulation of food intake in fish, and provide a new insight to understand the molecular mechanism of feeding regulation in Yangtze sturgeon during the weaning period.

## 2. Materials and Methods

### 2.1. Animals

The Yangtze sturgeon cultivated artificially for use in this study were obtained from the Fishery Institute of the Sichuan Academy of Agricultural Sciences (Yibin, Sichuan), and maintained in the fish tank (0.14 m^3^) at Sichuan Agricultural University. Continuously aerated and filtered water was supplied and the water temperature was maintained at 21 ± 3 °C under a natural light period. The Yangtze sturgeon were acclimated to a feeding schedule, namely: 14:00 once a day at the ratio of 3% body weight with a sturgeon commercial diet (Tongwei, Chengdu, China).

All animal experiments were approved by the Sichuan Agricultural University Animal Care and Use Committee with the approval numbers 2019202035-2106, 2019202035-2107, B20172101-1802, B20172101-1808, 2019202035-2108.

### 2.2. Experimental Fish and Procedures

For the cloning experiment and tissue distribution experiment, five juvenile Yangtze sturgeon (277.1 ± 31.1 g) were anesthetized with 0.01% MS-222, sacrificed and sampled at six hours before feeding. Tissues including the whole brain, esophagus, pyloric caeca, cardiac stomach, pyloric stomach, duodenum, valvula intestine, rectum, pancreas, liver, kidney and spleen were sampled, frozen into liquid nitrogen, and kept at −80 °C for RNA extraction.

During the weaning experiment, 180 juvenile fish at 60 days after hatching (dph) were divided into 18 groups (n = 10 per group). Among them, eight groups of fish were fed with *T. Limnodrilus* as control and sampled at 0, 1, 2, 3, 5, 6, 8 and 10 days. Seven groups of fish as weaning groups were provided with commercial microcapsule feed with crude protein ≥ 40%, crude fat ≤ 12%, crude fibre ≤ 6% and crude ash ≤ 18% (Tongwei Co., Ltd., Chengdu, China) after feeding with *T. Limnodrilus* for one day as food conversion treatment and sampled at 1, 2, 3, 5, 6, 8, and 10 days. Three groups of fish were refed with *T. Limnodrilus* after weaning and sampled at 6, 8 and 10 days. Five sturgeons were sampled from each group. After being anesthetized with 0.01% MS-222 and executed, the whole brain was dissected as described above and stored at −80 °C.

For the short-term fasting experiment, 63 fish (43.4 ± 6.8 g) were divided into seven groups (n = 9 each group) and fed at 2:00 p.m. for two weeks prior to the experiment. During the experiment, the fish were unfed at the scheduled time of feeding (14:00) and sampled at 6 h, 3 h and 1 h prior to feeding (−6 h, −3 h, −1 h), 0 h before feeding, 1 h, 3 h and 6 h after feeding (+1 h, +3 h and +6 h), with 0 h serving as the control. The tissues including the whole brain and duodenum were sampled, as described above.

For the intraperitoneal injection test, 72 healthy fish (56.3 ± 8.3 g) were divided into eight groups (three tanks each group, three fish each tank), and intraperitoneal injected with PBS, 3, 30 and 90 ng/g BW motilin-NH_2_, referenced to previous study [28,39]. The fish were provided with pre-weighted feeds at 0 h, 1 h and 3 h post injection with different solutions, and residual feeds were collected for the calculation of cumulative food intake to determine the effective dose. In addition, 36 fish (61.6 ± 8.6 g) were divided into four groups (n = 3 × 3/group), intraperitoneal injected with PBS and an effective dose of motilin-NH_2_ (90 ng/g BW). Six fish from two groups were selected to calculate their digestive tract filling index (filling index = weight of food clumps in digestive tract ×10,000/weight of the fish after it has been gutted). The whole brain and stomach from the other two groups were collected at 1 h post-injection, as described above, to analyse the mRNA expression of appetite factors.

The in vitro incubation of the whole brain and stomach fragments was conducted following the methods previously described [34]. *Acipenser dabryanu* (n = 9, 679.1 ± 101.7) were sacrificed, and then fresh whole brain and stomach were collected. After washing with PBS three times, all the fragments were cut into nearly 1 mm^3^. The fragments were evenly distributed into a 24-well plate containing 1 mL DMEM medium and 10% fetal bovine serum (FBS). After preincubation for 3 h at 25 °C, different doses of motilin-NH_2_ peptide (10^−6^ M and 10^−7^ M) and PBS were added into the marked test wells. Each treatment containing three replicates was kept at 25 °C for 1 h and 3 h. Tissues were collected into a centrifuge tube and quickly frozen in liquid nitrogen for the extraction of RNA and real-time PCR.

### 2.3. Molecular Cloning and Sequence Analysis of Motilin and Motilin Receptor

To obtain *Acipenser dabryanu motilin* (*AdMotilin*) and *motilin receptor* (*AdMotilinR*), the core fragment primers, 5′ RACE primers and 3′ RACE primers (Table 1) of *motilin* and *motilin receptor* were designed with Software Primer 5.0 based on transcriptome database of *Acipenser dabryanu*. PCR amplification was performed with 10 μL volume including 2× Taq PCR PreMix (5 μL; Takara, Dalian, China), ddH_2_O (3 μL), cDNA sample (1 μL), forward primer (0.5 μL) and reverse primer (0.5 μL), respectively. The PCR program includes: denaturation at 94 °C for 5 min, followed by 35 cycles of 94 °C for 30 s, annealing for 30 s, 72 °C for 1 min, and final incubation for 5 min at 72 °C. The products were purified with an agarose gel DNA recovery kit (TIANGEN, Beijing, China) and cloned into pMD19-T (Takara, Dalian, China) for sequencing.

The sequences of *motilin* and *motilin receptor* genes predicted ORF using EditSeq and analyzed the homology by BLAST. The signal peptide of *AdMotilin* and *AdMotilinR* were predicted using SignalP 4.0 Server (http://www.cbs.dtu.dk/services/SignalP/, accessed on 26 March 2024). The alignments of multiple sequences were performed by DNAMAN software combined with BOXSHADE 3.2 (http://www.ebi.ac.uk/Tools/msa/clustalw2/, accessed on 26 March 2024). The protein structure was predicted using I-TASSER and the structural domains of which were predicted with TMHMM Server, v. 2.0. The construction of the phylogenetic tree was carried out with MEGA 7 software (https://www.megasoftware.net/, accessed on 26 March 2024).

### 2.4. Reagents

The Yangtze sturgeon motilin-NH_2_ (FLSFFSPSDMRRMMEKEKSKTG-NH_2_) was synthesized by Shanghai Bootech BioScience & Technology Co., Ltd. (Bootech, Shanghai, China). The purity was >95%, which was determined with analytical HPLC. The peptides were dissolved in the PBS and stored at −20 °C.

### 2.5. Quantitative Real-Time PCR

Total RNA was extracted from the target tissues using RNA Isolation Kit (Forgene, Chengdu, China). The concentration and density of RNA were determined using electrophoresis and a NanoDrop 2000 spectrophotometer (Thermo Scientific, Waltham, MA, USA). RNA of 1 μg was reversed into cDNA with PrimeScript^TM^ RT Reagent Kit 047A (TaKaRa, Dalian, China) for qPCR reaction.

The specific primers of target genes were designed using Primer 5 software (Table 1). The reference genes including *β-actin* and *ef1-α* were used to normalize the gene expression based upon previous study in *Acipenser dabryanu* [40]. The qRT-PCR was performed on CFX96 real-time PCR system containing 2 × Taq SYBR Green qPCR Premix (5 μL), dd H_2_O (3.2 μL), sample cDNA (1 μL), forward primer (0.4 μL) and reverse primer (0.4 μL), respectively. The qPCR program was as follows: 95 °C for 2 min, 95 °C for 5 s and annealing for 30 s in 39 cycles. The expression levels of target genes were quantified with the 2^−∆∆CT^ method.

### 2.6. Statistical Analysis

Data were presented as mean ± SEM. The data were analyzed by Student’s *t*-tests (two groups) after testing the normality. One-way ANOVA followed by LSD post hoc test (multiple groups) with SPSS 21.0 version after testing the homogeneity of variances. *p* < 0.05 was considered to be statistically significant.

## 3. Results

### 3.1. Molecular Cloning of AdMotilin and AdMotilinR

The *AdMotilin* partial sequence of 561 bp was cloned (GenBank:OM141116) containing 5′-UTR of 38 bp, 3′-UTR of 184 bp and ORF of 339 bp. The *AdMotilin* ORF encoded 112 amino acids (aa), which can be cleaved into a signal peptide of 25 aa, mature peptide of 22 aa, and motilin-related peptides (Figure 1A). *AdMotilinR* cDNA of 1041 bp was obtained (GenBank:OM141117) which encoded 346 amino acids containing seven transmembrane domains (Figure 1B).

The amino acid structure and mature peptide restriction site KK of *AdMotilin* were different from those of other fish, but consistent with those of mammals and birds (Figure 2A). The motilin receptor of all species is highly conservative in the TM region, but compared with mammals, there is about 40 bp loss between TM4 and TM5 in birds, reptiles, amphibians and fish (Figure 2B). Amino acid identity analysis showed that the motilin-coding region of Yangtze sturgeon was 54.6% consistent with that of reedfish (*Erpetoichthys calabaricus*), and 36.1–45.4% consistent with that of humans (*Homo sapiens*), chicken (*Gallus gallus*), zebrafish (*Danio rerio*) and spotted sea bass (*Lateolabrax maculatus*) (Figure 2C). Additionally, the mature peptide of *AdMotilin* was 75% consistent with that of reedfish, and the identity of *AdMotilin* between humans, spotted sea bass, chicken and zebrafish were 38.1%, 47.1%, 47.6% and 52.9%, respectively (Figure 2C).

The results of the I-T ASSER analysis showed the similarity of the protein structure of motilin in vertebrates (Figure 3). The phylogenetic tree showed that the *Acipenser dabryanus* motilin sequence was clustered with other fish, and closely related to the snakefish (Figure 4A). The phylogenetic tree shows that the *Acipenser dabryanus* motilin receptor is clustered with other fish and closely related to the motilin receptor predicted in sterlet (*Acipenser ruthenus*) and snakefish (Figure 4B).

### 3.2. Tissue Distribution

The mRNA of *AdMotilin* was detected in several tissues of *Acipenser dabryanus.* The mRNA level of *AdMotilin* was predominantly expressed in the whole brain, followed by the pyloric caeca, duodenum, valvula intestine and rectum, and low level of *AdMotilin* mRNA was detected in the liver, spleen, pyloric stomach, esophagus, pancreas, cardia stomach and kidney (*p* < 0.01, Figure 5A). The mRNA of *AdMotilinR* was abundantly expressed in the cardia stomach, followed by the duodenum, pyloric stomach, whole brain, valvula intestine and rectum, but was lower expressed in the pancreas, pyloric caeca, esophagus, spleen, liver and kidney (*p* < 0.01, Figure 5B).

### 3.3. Effects of Weaning on the Expression of AdMotilin and AdMotilinR

To explore the potential role of the motilin system during the weaning period of Yangtze sturgeon juveniles, *AdMotilin and AdMotilinR* expression in the brain were detected. Compared with the control group fed with *T. Limnodrilus*, *AdMotilin* mRNA expression in the brain of the weaning group, which was provided with commercial feed, was significantly increased from day 1 to day 10. When the fish in the weaning group were refed with *T. Limnodrilus*, *AdMotilin* expression was significantly decreased on day 6, day 8 and day 10, respectively(Figure 6A). The expression of *AdMotilinR* of the brain in the weaning group increased significantly on day 3, day 5 and day 6 (*p* < 0.01) (Figure 6B). Relative to the weaning group, the expression of *AdMotilinR* in the brain of fish refed with *T. Limnodrilus* was not changed immediately but significantly increased on 8d (*p* < 0.01).

### 3.4. Effects of Fasting on AdMotilin and AdMotilinR Expression

To explore the potential function of *AdMotilin* and *AdMotilinR* in the brain and duodenum, the expression of these genes was determined after fasting for several hours in *Acipenser dabryanus. AdMotilin* mRNA expression in the brain was significantly increased along with the fasting time, with higher expressions at 3 h and 6 h after fasting (Figure 7A). By contrast, *AdMotilin* expression in the duodenum was significantly decreased with the extension of fasting time, and then reached the lowest expression level at 6 h (*p* < 0.01, Figure 7B). In addition, *AdMotilinR* expression in the brain was significantly increased and reached the highest level at 1 h after fasting (*p* < 0.05), and then *AdMotilinR* expression level returned to the level at 0 h after 3–6 h fasting (Figure 7C). Compared with 0 h, no significant difference was observed in the expression of *AdMotilinR* expression in duodenum after the fasting for 1–6 h (Figure 7D). Before the scheduled feeding time (0 h), the mRNA expression of *motilin* in the brain was increased at −1 h (Figure 7A) and *motilinR* expression in the duodenum was increased at −3 h and −1 h (Figure 7D).

### 3.5. Effects of Motilin-NH_2_ Injection on Food Intake of Yangtze Sturgeon

Intraperitoneal injection of 3 ng/g BW motilin-NH_2_ didn’t significantly affect the food intake, 30 ng/g BW motilin-NH_2_ significantly increased the 1–3 h food intake and 3–6 h food intake, and 90 ng/g BW motilin-NH_2_ promoted 0–1 h and 1–3 h food intake (*p* < 0.05) (Figure 8A). The cumulative food intake at 1 h, 3 h and 6 h post-administration was significantly increased by motilin-NH_2_ (90 ng/g BW), but 30 ng/g BW motilin-NH_2_ only significantly increased at 6 h cumulative food intake when compared with *i.p.* injected with PBS (Figure 8B). In addition, the digestive tract filling index was detected after *i.p.* injection of 90 ng/g BW motilin-NH_2_, which showed a significant increase when compared with the control group (Figure 8C).

### 3.6. Effects of Motilin-NH_2_ Injection on AdMotilinR Expression

To explore whether the motilin receptor was involved in the feeding control by exogenous motilin-NH_2_, *AdMotilinR* expressions in the brain and stomach were detected at 1 h after injection. When fish were treated with 90 ng/g BW motilin-NH_2_, the expressions of *AdMotilinR* in the brain (*p* < 0.05, Figure 9A) and stomach (*p* < 0.01, Figure 9B) were significantly decreased.

### 3.7. Effects of Motilin-NH_2_ Injection on Appetite Factors Expression

To elucidate the appetite mechanism of motilin-NH_2_ in feeding control of Yangtze sturgeon, the expression of several appetite factors in the brain (*POMC*, *CART*, *NPY*, *AgRP*, *Apelin*, *NUCB2*, *NPFF*, *Ghrelin*, *CCK* and *Gastrin*) and stomach (*Ghrelin*, *CCK* and *Gastrin*) were examined after intraperitoneal injection of 90 ng/g BW motilin-NH_2_. Regarding the expression of appetite factors in the brain, the levels of *POMC*, *CART*, *NPY*, *AgRP*, *Apelin*, *NUCB2* and *CCK* were significantly decreased, however, the expression level of *NPFF*, *Ghrelin* and *Gastrin* were not significantly changed (Figure 10A). In the stomach, the decrease of *CCK* mRNA expression was noted after 1 h post-injection of motilin-NH_2_ (*p* < 0.01), whereas, the expression of *Ghrelin* and *Gastrin* expressions showed no significant change (Figure 10B).

### 3.8. In Vitro Effects of Motilin-NH_2_ Incubation on Appetite Factors Expression in the Brain and Stomach Fragments

To explore the possible roles of motilin on Yangtze sturgeon, in vitro experiments of motilin on the expression of appetite factors in the brain fragments and stomach fragments were conducted. After the incubation with 10^−6^ M motilin-NH_2_, *POMC* mRNA expression was significantly inhibited at 1 h post-treatment (*p* < 0.01), while 10^−7^ M and 10^−6^ M motilin-NH_2_ significantly increased *POMC* mRNA expression at 3 h post-treatment when compared with the PBS control group (*p* < 0.01, *p* < 0.05, Figure 11A). The expression of *CART* was significantly inhibited at 1 h after the incubation with motilin-NH_2_ (10^−7^ M and 10^−6^ M), but increased at 3 h after treatment with 10^−6^ M motilin-NH_2_ (*p* < 0.05, Figure 11B). Notably, the expression of *NPY* was significantly increased at 1 h and 3 h after 10^−7^ M and 10^−6^ M motilin-NH_2_ administration (Figure 11C). Similarly, both the 10^−7^ M and 10^−6^ M motilin-NH_2_ incubation for 1 h significantly promoted *AgRP* expression (*p* < 0.05, *p* < 0.01), but did not affect the expression of *AgRP* at 3 h post-treatment (Figure 11D).

In the brain, the 10^−7^ M and 10^−6^ M motilin-NH_2_ significantly inhibited the expression of *CCK* at 1 h (Figure 12A). Similarly, the expression of *CCK* was decreased after incubation with 10^−6^ M motilin-NH_2_ for 1 h in the stomach, and the two concentrations of motilin-NH_2_ significantly inhibited the expression of *CCK* at 3 h (Figure 12B).

## 4. Discussion

### 4.1. The Analysis of AdMotilin and AdMotilinR Sequences in Yangtze Sturgeon

In mammals, motilin and the motilin receptors have been well-studied [14]. However, the knowledge about motilin and the motilin receptor in fish is limited. In teleost, motilin and the motilin receptor have been reported in zebrafish [31] and spotted sea bass [34]. In this study, Yangtze sturgeon motilin (*AdMotilin*) was obtained for the first time. The amino acid sequence analysis showed that amino acids encoded by *AdMotilin* were low when compared with other fish, chickens and humans (31.4–54.3%) [16,19,31,34]. The Yangtze sturgeon motilin precursor comprised a signal peptide, which was similar to other vertebrates, suggesting Yangtze sturgeon motilin may serve as a secretory protein. A shear site (KK) of AdMotilin located between the motilin mature peptide (MLN) and the motilin associated protein (MAP), which has been found in humans, pigs, cows, rabbits and chickens [19,20,41], and a shear site RK in the corresponding regions was observed in zebra fish [31] and spotted sea bass [34]. Yangtze sturgeon motilin mature peptides (MLN) showed a relatively low identity with zebra fish (52.9%), chicken (47.6%), spotted sea bass (47.1%), and human (38.1%), suggesting Yangtze sturgeon motilin mature peptide was not conservative. Additionally, the mature peptide of AdMotilin showed relative higher amino acids identity with that of reedfish (75%) which is a representative species of non-teleost actinopterygian fish groups like the sturgeon [42].

Recently, several studies showed that *motilin* and *motilin receptor* were gradually degraded [43], absent [44] or existed in the form of pseudo-genes [45] in some rodents, while mature peptides of motilin from other species could still perform biological functions in rats [46] and shrews [47]. As a primitive and ancient fish, Yangtze sturgeon still has complete coding regions of *motilin* and *motilin receptor*. However, sturgeon has undergone polyploidization after several genome-wide replication events, showing a special evolutionary status [48], which might be part of the reasons why the amino acids encoded by AdMotilin were low in compared to other species.

Several studies in mammals [49,50], birds, and fish including zebra fish and spotted sea bass [31,34] have identified the motilin receptor. Two selective shear forms of motilin receptor (GPR38-A and GPR38-B) were reported in humans, of which the GPR38-A contains seven transmembrane domains with a higher affinity to motilin whereas the GPR38-B only contains the first five transmembrane structures of GPR38-A [51]. In this study, only one motilin receptor composed of seven transmembrane domains has been cloned from the Yangtze sturgeon, and the result was similar to that of GPR38-A. Whether AdMotilinR contains another shear form of motilin receptor remains to be determined.

### 4.2. The Tissue Expression of AdMotilin and AdMotilinR

To investigate the roles of *AdMotilin* and *AdMotilinR*, this study detected their tissue distribution in juvenile Yangtze sturgeon using RT-qPCR. *Motilin* and *motilin receptor* were widely expressed in the brain and peripheral tissues in mammals. In the present study, the results showed that *AdMotilin* mRNA was highly expressed in the brain of the Yangtze sturgeon. Inconsistent with this study, motilin was reported to be mainly distributed in the duodenum of birds such as Japanese quail [20] and mammals such as rabbit [16], guinea pig [41], stinking shrew [18]. In addition, in other fish, *motilin* was not only highly expressed in the intestine, but also in the liver of zebra fish [31] and the spleen of spotted sea bass [34]. These observations suggest that motilin might have multifunctional roles in fish, therefore the function of motilin in the brain of Yangtze sturgeon deserves attention.

*AdMotilin* and *AdMotilinR* were abundantly expressed in the gastrointestinal tract of Yangtze sturgeon, which is similar to that of mammals and birds with abundant expression of motilin in the stomach and duodenum [52,53,54]. However, the tissue expression of *AdMotilinR* was distinct from that of other fish species. For instance, in zebra fish, the *motilin receptor* was highly expressed in the optic tectum thalamus, hypothalamus, and hindbrain [31]. In spotted sea bass, the *motilin receptor* was abundantly expressed in the intestine and pronephros, but lower expression was noted in the brain [34]. In all, these results indicate that the mRNA expression of *motilinR* in tissues is species-specific. High levels of *motilin* and *motilin receptor* were observed in the brain and gastrointestinal tract, suggesting that these tissues might be the key sites for motilin function in Yangtze sturgeon, which needs to be further studied.

### 4.3. Weaning and Fasting Induced AdMotilin and AdMotilinR Expression Changes

Weaning is a crucial stage during the early life of fish, and many fish have difficulty in food conversion from live bait to artificial feed. In sturgeons, it has been reported that the change in weaning strategies such as the start time, diet selection, and feeding methods affect the survival, growth and metabolism in white sturgeon [55], sterlet [56] and Yangtze sturgeon [57]. The mechanisms underlying the decrease in food intake induced by weaned diets are unclear in fish, which might be controlled by hormones secreted from the brain [58]. This present study reports the change of *motilin* and *motilin receptor* mRNA expressions in response to the weaning. The results show that the expression of *AdMotilin* in the brain increased during the weaning process, and refeeding inhibited the up-regulation of *AdMotilin* expression caused by weaning. Similarly, a previous study in flatfish has shown that orexigenic factor *ghrelin* expression was increased gradually with the extension of weaning time [59]. Additionally, a previous study on Yangtze sturgeon has reported that anorexigenic factor *leptin* expression in the weaning group was lower than un-weaned fish for the first five days [10], which was contrary to our results. In grass carp, when fish were fed with plant protein to replace fish meal, the expressions of orexigenic factors (*Ghrelin* and *AgRP*) were up-regulated and anorexigenic genes (*Leptin*, *Pomc* and *Pyy*) expressions were down-regulated [60]. These results showed that the expression of appetite factors were closely related to whether the obtained food is delicious during the process of weaning, and hence, motilin may serve as an indicator for food conversion. However, by contrast to the regular change of *motilin* expression induced by weaning, the expression of *motilinR* in the brain of Yangtze sturgeon varies in a complicated way during the weaning time, which might be due to the fact that this study detected the mRNA expression of *motilin* and *motilinR* and a delayed response exists between *motilinR* and *motilin*.

The change of feeding status is closely associated with the expression of appetite factors in the blood and tissues [10]. To investigate the potential role of motilin and motilin receptor in feeding of Yangtze sturgeon, we detected *AdMotilin* and *AdMotilinR* expressions in the duodenum and brain under fasting. The results showed that *AdMotilin* mRNA expression in the duodenum was up-regulated after fasting for 3 h but down-regulated after fasting for 6 h. Similarly, in the intestine of spotted sea bass, the expression of *motilin* reached its highest level in 1 h after fasting and the lowest expression in 6 h after fasting [34]. Previously, in human, the concentration of plasma motilin was positively correlated with the hunger score, which were increased significantly after fasting for 45 min and 12 h [27,61]. Previous studies have reported that the expression of many gastrointestinal hormones is stimulated after a meal [62,63]; however, the expression of *AdMotilin* in the intestine increased after fasting. By contrast with other studies, this study found that *AdMotilin* was mainly expressed in Yangtze sturgeon brain and its expression was increased after fasting for 3 h and 6 h. Meanwhile, *AdMotilinR* expression in the brain was promoted at 1 h after fasting, which was not examined in the duodenum of Yangtze sturgeon. These results suggest that the increase of *AdMotilin* expression may serve as a hunger indicator during weaning and food deficiency, and further investigation is needed to clarify the role of motilin in the regulation of feeding. Several appetite factors such as ghrelin have been reported to be an orexigenic signal, namely preprandially rising. Interestingly, in this study, the mRNA expression of AdMotilin in the brain was increased at 1 h before the scheduled feeding time, suggesting motilin may act as a driving factor in promoting appetite behavior in the Yangtze sturgeon. It is noted that the increase of MotlinR expression in the duodenum before the feeding time occurred earlier than motilin mRNA expression, suggesting motilinR might be regulated by nutritional status but not motilin. However, this study only detected mRNA expression of motilin and motilinR. It needs to be noted that the change of expression in transcript does not necessarily result in the change of protein expression. Further studies should be conducted using sturgeon derived specific antibody to verify if the protein levels of AdMotilin and AdMotilinR in the brain and duodenum are altered by different nutritional status.

### 4.4. Motilin Polypeptide Promoted Food Intake of Yangtze Sturgeon

The existence of periprandial change of *AdMotilin* led us to further explore whether AdMotilin regulates the appetite of Yangtze sturgeon. Our results showed that motilin polypeptide (motilin-NH_2_) intraperitoneal injection significantly promoted food intake and the digestive tract filling index of Yangtze sturgeon. Consistent with our results, a previous study in rats showed that motilin increased food consumption at 15 min, 1 h and 2 h in a time-dependent manner [28]. Similarly, intravenous infusion of motilin significantly increased hunger scores and food intake in humans [27,64]. Meanwhile, intraventricular injection of motilin promoted feeding of rats [28,39,46] and mice [65]. These results indicate that AdMotilin can promote food intake of Yangtze sturgeon. Notably, the gastrointestinal filling index of Yangtze sturgeon was significantly up-regulated after intraperitoneal injection of motilin-NH_2_, suggesting motilin may promote food intake by acting on the gastrointestine.

To explore the mechanism of motilin-induced orexgenic effect in Yangtze sturgeon, *AdMotilinR* expressions in the stomach and brain were determined by RT-qPCR after intraperitoneal injection of motilin-NH_2_ (90 ng/g BW). Previous studies in mice and rats have shown that the increase in food intake induced by motilin at 20 min, 1 h and 2 h could be reversed after motilin receptor antagonist (GM-109) administration [46,65]. In fish, only one previous study in zebrafish has revealed the potential interaction between motilin and motilin receptor. In the COS-7 cells, the motilin-like peptide enhanced the CRE- and SRE-promoter that activates the motilin receptor of zebrafish [31]. However, whether the motilin receptor is involved in the regulation of feeding induced by motilin in fish is not clear. Our results showed that *AdMotilinR* in the brain and stomach might participate in the change of food intake induced by intraperitoneal injection of motilin-NH_2_ in the Yangtze sturgeon. Further study needs to explore the role of AdMotilinR in mediating the feeding stimulation induced by motilin using receptor antagonists or specific interfering RNAs in Yangtze sturgeon.

Food intake is mainly controlled by related signals transmitted by other neuropeptides (such as AgRP, NPY, CART, POMC, NUCB2, Apelin) and gastrointestinal hormones (such as ghrelin, CCK) and integrated by NPY/AgRP neurons and POMC/CART neurons located in the central nervous system [66,67]. In fish, current studies have shown that *NPY*, *AgRP* and *Ghrelin* can promote feeding [7,68,69,70], while *POMC*, *NUCB2* and *CCK* can inhibit feeding [66,71,72]. In sturgeon, *CART* and *Apelin* have bidirectional effects including anorexia action in the short term [73,74]. We observed that intraperitoneal injection of motilin-NH_2_ reduced appetite factors *POMC*, *CART*, *Apelin*, *NUCB2* and *CCK* expressions in the brain of Yangtze sturgeon. These results are consistent with the role of these anorexigenic factors in feeding regulation of Atlantic salmon [75], Siberian sturgeon [73,74] and Yangtze sturgeon [76]. Similarly, the results of the in vitro experiment showed that *POMC*, *CART* and CCK mRNA expressions in the brain were decreased after treatment with 10^−6^ M motilin-NH_2_ for 1 h. However, these inhibitory effects in the brain were reversed at 3 h after motilin peptide treatment in vitro, suggesting motilin-induced appetite may be inhibiting the expression and release of anorexia factors including *POMC*, *CART* and *CCK* in the brain, although this effect is rapid but not persistent. Whereas we have observed the decrease of *NPY* and *AgRP* expression in the brain at 1 h after intraperitoneal injection of motilin-NH_2_ peptide which is opposite to the orexigenic roles of *NPY* and *AgRP* in mice [68], zebrafish [70], rainbow trout [77], tilapia [78], Atlantic salmon [79], and Siberian sturgeon [7]. In contract, incubation with motilin-NH_2_ significantly elevated NPY and AgRP expression in brain fragments, and the stimulating effect on NPY expression was persistent. This difference might be due to the fact that intraperitoneal injection is a peripheral drug delivery method that acts on the brain indirectly, and the appetite factors produced from peripheral tissues may have a negative feedback effect on the expression of NPY and AgRP in the brain. In addition, tissue incubation of motilin directly affects the expression of NPY and AgRP in different brain regions. These results suggest that the effect of peripheral motilin-NH_2_ on increasing food intake may not be directly mediated by NPY and AgRP in the central nervous system, and the different effect of central motilin administration on appetite regulation needs to be further investigated. Furthermore, intraperitoneal injection or incubation of motilin-NH_2_ induced the decrease of *CCK* expression in the stomach. A different result was observed in the intestine of spotted sea bass by motilin after 3-h treatment [34]. This result suggests that motilin regulate the expression of *CCK* in a species- and tissue-specific manner. This difference might be due to the physiological differences of CCK in promoting movement in the gut, reducing gastric pressure and inhibiting gastric emptying in the stomach [12]. In humans, motilin can act as physiologic regulator of gastric phase III contractile activity which is an important determinant of the return of hunger after a meal [27,64]. These results suggest that the orexigenic role of motilin may be mediated by CCK in the stomach and brain. As the interaction between appetite factors is complicated in the brain, we hypothesized that motilin may interact with other anorexigenic factors and orexigenic factors to promote food intake through direct or indirect mechanisms, and further studies are needed to explore the effect on different brain regions, such as homeostatic or hedonic regions, in response to motilin.

## 5. Conclusions

Taken together, the present study isolated the *motilin* and *motilin receptor* gene from Yangtze sturgeon, and the motilin mature peptide consisted of 22 amino acids with low identity with other reported species including mammals, birds and fish. It was found that *AdMotilin* was predominantly expressed in the brain of Yangtze sturgeon, and also the *AdMotilinR* was highly expressed in the gastrointestinal tract. The increased expression of *motilin* mRNA in the brain of the weaned fish and fasted fish suggested a hunger signal when food is palatable or limited. Further, the results obtained in this present study provide direct evidence for the food-promoting effect of motilin in the Yangtze sturgeon, which could be related to the inhibition of the expression of *POMC*, *CART* and *CCK* genes in the Yangtze sturgeon. In addition, we found that the in vitro motilin-NH_2_ peptide incubation for 1 h promoted the expression levels of *AGRP* and *NPY* in the brain. Therefore, these observations provide a better understanding of the regulatory roles of motilin on food intake in fish. In addition, the brain mechanisms underlying the stimulatory effect of motilin in food intake should focus on different brain regions such as hypothalamus, telencephalon and midbrain in appetite regulation. However, further studies are needed to explore the application of motilin peptide during weaning in fish production.

## Figures and Tables

**Figure 1 biomolecules-14-00433-f001:**
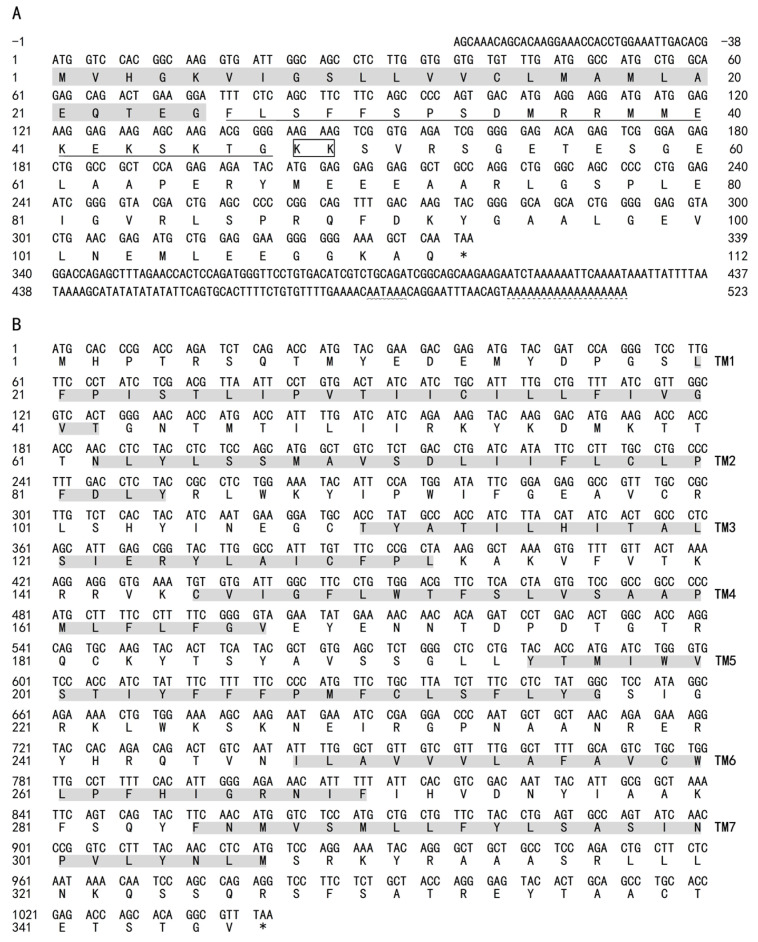
The cDNA sequence and deduced amino acids of AdMotilin (**A**) and AdMotilinR (**B**). (**A**) Shadow: signal peptide, transmembrane domain; Underline: mature peptide; Black box: enzyme restriction site; Wavy line: tail modifier; Dashed line: tails of poly A. (**B**) Shadow: transmembrane domain. The asterisk represents the termination codon.

**Figure 2 biomolecules-14-00433-f002:**
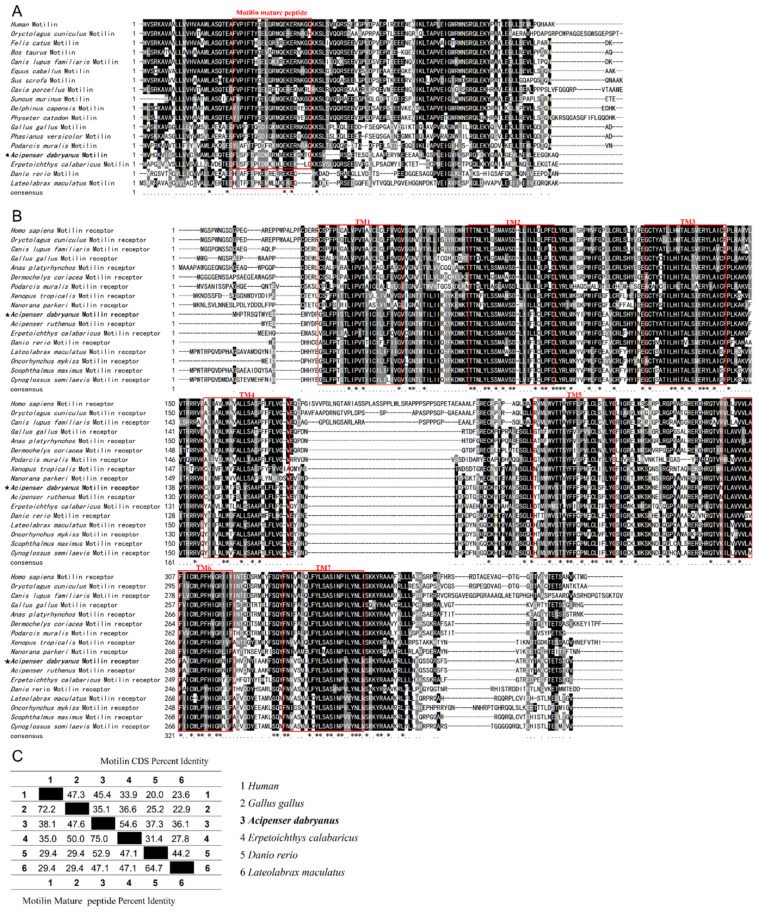
Amino acid sequence alignment between motilin (**A**) and motilin receptor (**B**), consistency of motilin CDS and mature peptide (**C**). The symbol of the five pointed star represents the sequence of Yangtze sturgeon. The asterisk represents completely identical amino acids. Black shadows represent the same amino acids, while gray shadows represent similar amino acids (consistency exceeding 60%).

**Figure 3 biomolecules-14-00433-f003:**
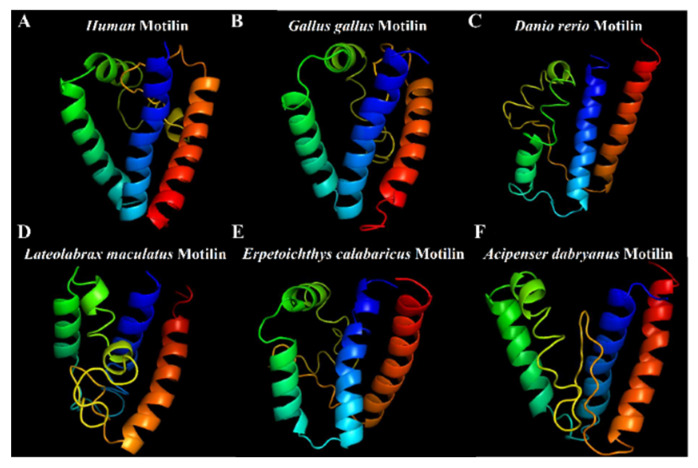
Protein structure of motilin. Human (*Homo sapiens*) (**A**); chicken (*Gallus gallus*) (**B**); zebrafish *Danio reio* (**C**); Chinese seabass (*Lateolabrax maculatus*) (**D**); snakefish (*Erpetoichthys calabaricus*) (**E**) and Yangtze sturgeon (*Acipenser dabryanus*) (**F**). Blue, red and green represent three alpha helixs, and yellow means random coil.

**Figure 4 biomolecules-14-00433-f004:**
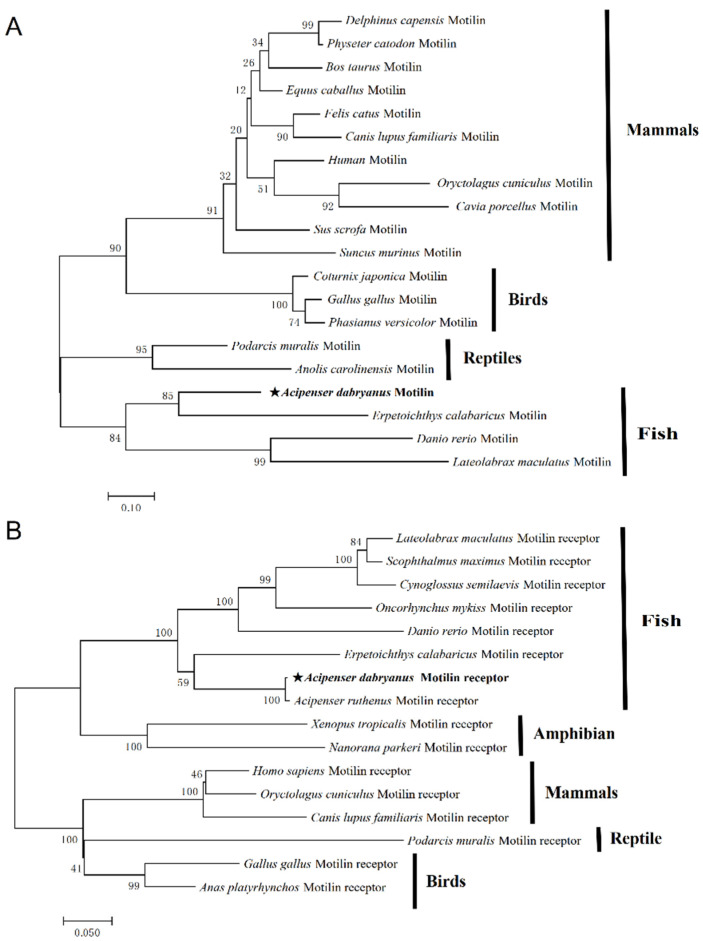
Phylogenetic tree analysis of motilin (**A**), and motilin receptor (**B**). The asterisk represents the sequence of Yangtze sturgeon.

**Figure 5 biomolecules-14-00433-f005:**
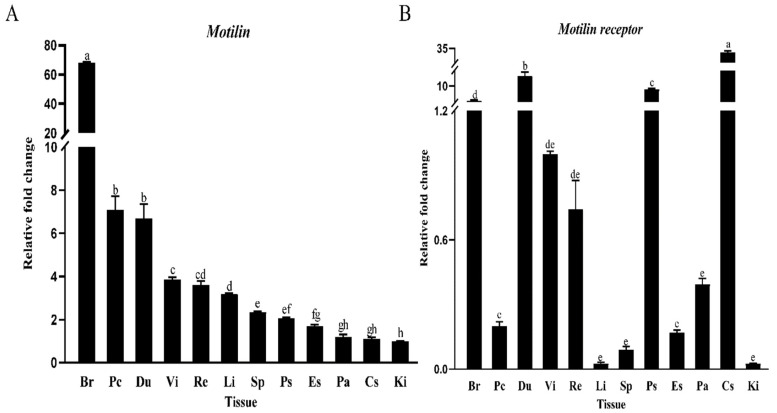
Tissue distribution of *AdMotilin* (**A**), and *AdMotilinR* (**B**). Br, Brain; Pc, Pyloric caeca; Du, Duodenum; Vi, Valve intestine; Re, Rectum; Li, Liver; Sp, Spleen; Ps, Pyloric stomach; Es, Esophagus; Pa, Pancreatic; Cs, Cardia stomach; Ki: Kidney. Different letters represent significant differences in mRNA expression among tissues (*p* < 0.05).

**Figure 6 biomolecules-14-00433-f006:**
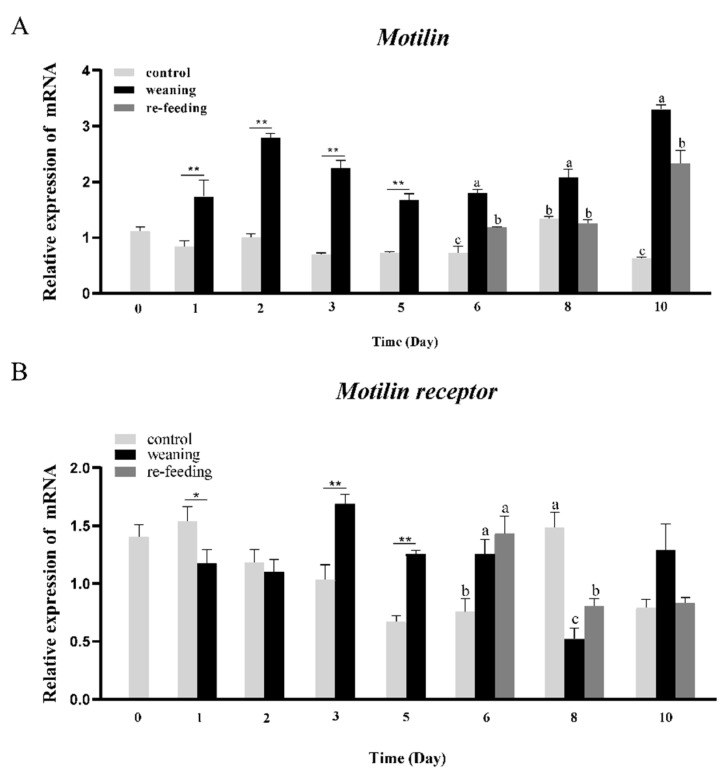
Weaning induces changes in *AdMotilin* (**A**) and *AdMotilinR* (**B**) expression in the brain of Yangtze sturgeon. Different letters represent significant differences among the groups. Asterisks represent significant differences between the control group and the weaning group at the same time point (Student’s *t*-test, * *p* < 0.05 and ** *p* < 0.01).

**Figure 7 biomolecules-14-00433-f007:**
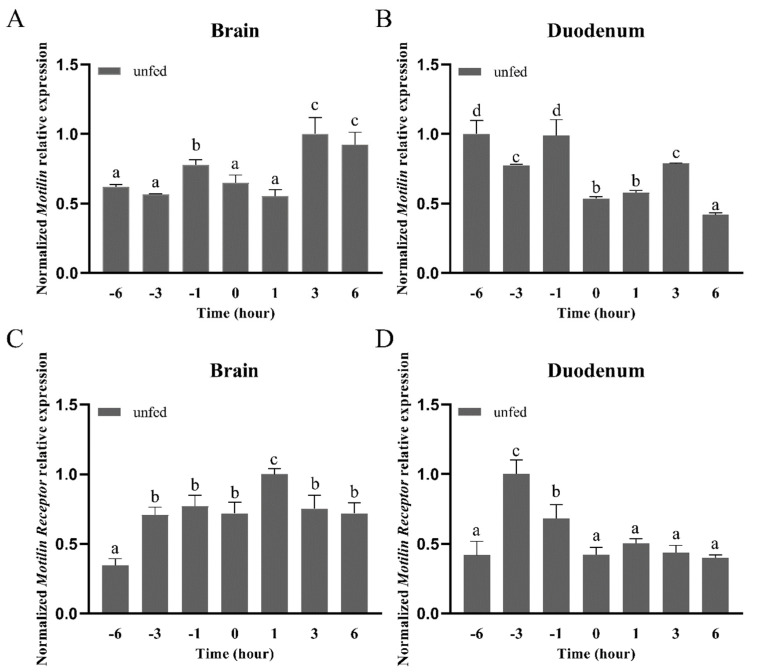
Short-term fasting induced changes in the expression of *AdMotilin* (**A**,**B**) and *AdMotilinR* (**C**,**D**) in the brain and duodenum. Different letters represent significant differences in mRNA expression among different time point (*p* < 0.05).

**Figure 8 biomolecules-14-00433-f008:**
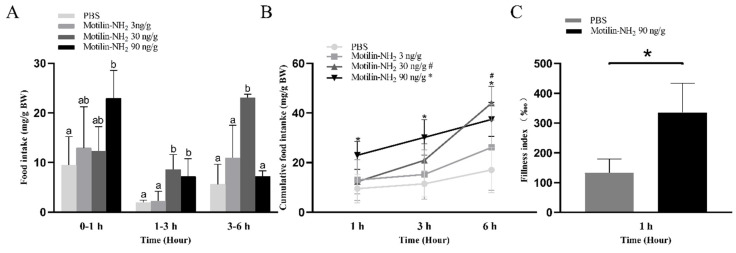
Effects of motilin-NH_2_ injection on food intake (**A**), cumulative food intake (**B**) and gastrointestinal filling index (**C**) of Yangtze sturgeon. # and * indicates a significant difference between intraperitoneal injection of motilin-NH_2_ (30 ng/g BW or 90 ng/g BW) and PBS group at the same time point, respectively. Different letters represent significant differences in food intake among different doses of drugs administration (*p* < 0.05).

**Figure 9 biomolecules-14-00433-f009:**
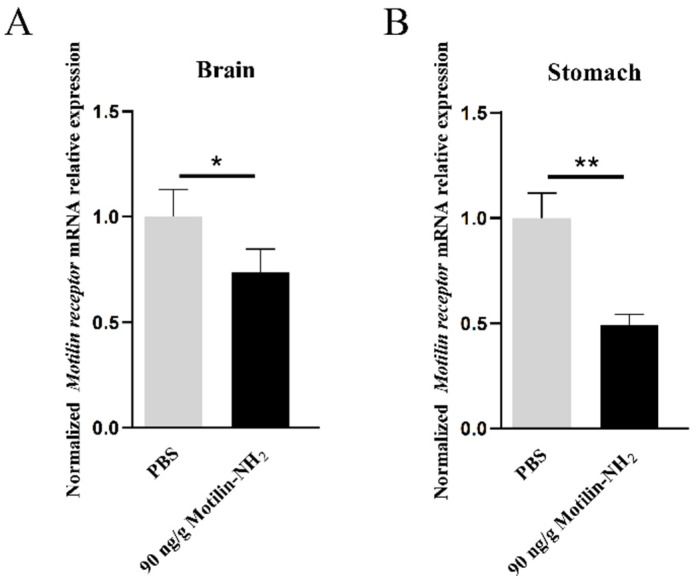
Change of *AdMotilinR* expression in brain (**A**) and stomach (**B**) after 90 ng/g BW motilin-NH_2_ injection_._ The asterisks indicate significant difference between the motilin-NH_2_ treatment group and the PBS control group. * represents *p* < 0.05, ** represents *p* < 0.01.

**Figure 10 biomolecules-14-00433-f010:**
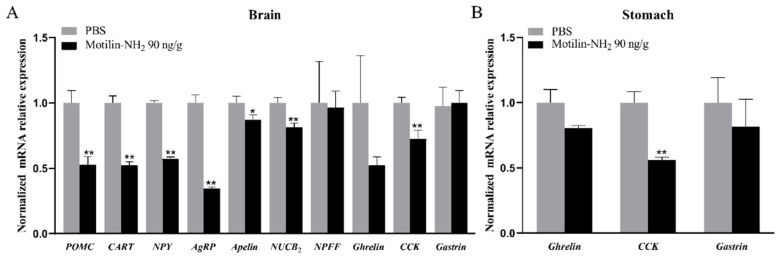
Effects of intraperitoneal injection of 90 ng/gBW motilin-NH_2_ on appetite factors expressions in the brain (**A**), and stomach (**B**) of Yangtze sturgeon. The asterisks indicate significant difference between the Motilinmotilin-NH_2_ treatment group and the PBS control group. * represents *p* < 0.05, ** represents *p* < 0.01.

**Figure 11 biomolecules-14-00433-f011:**
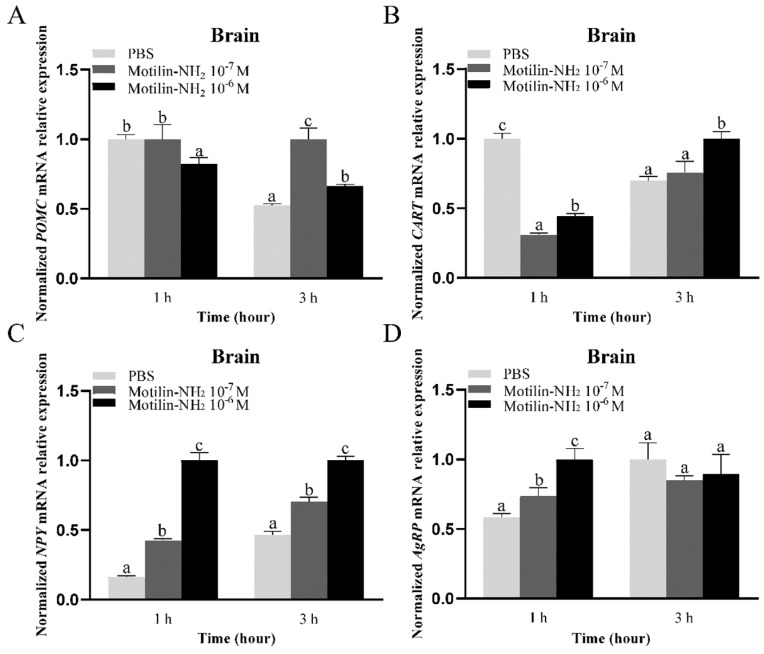
In vitro effects of motilin-NH_2_ on *POMC* (**A**), *CART* (**B**), *NPY* (**C**) and *AgRP* (**D**) expressions in the brain of Yangtze sturgeon. Different letters represent significant differences in mRNA expression among different doses of drugs administration (*p* < 0.05).

**Figure 12 biomolecules-14-00433-f012:**
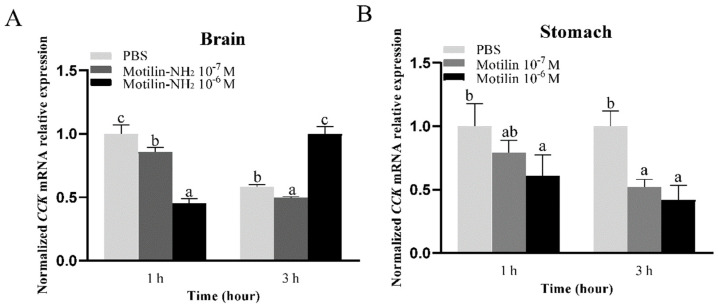
Effects of 10^−7^ M and 10^−6^ motilin-NH_2_ on the expression of CCK in the brain (**A**), and stomach (**B**) of Yangtze sturgeon. Different letters represent significant differences in mRNA expression among different doses of drugs administration (*p* < 0.05).

**Table 1 biomolecules-14-00433-t001:** The sequence of primers used in this study.

Primer Name	Sequences (5′–3′)	TM (°C)	Useage
*Motilin*-F	AGCAAACAGCACAAGGAAAC	56	Clone
*Motilin*-R	ATGTATCTCTCTGGAGCGGC		
*Motilin*-F1	GACACGATGGTCCACGGCAAGGT	62	3′ RACE
*Motilin*-F2	GATGGAGAAGGAGAAGAGCAAGACGG		
*Motilin receptor*-F	ATGCACCCGACCAGATCTCAGACC	68	Clone
*Motilin receptor*-R	TTAAACGCCTGTGCTGGTCTCGGT		
*Motilin*-qF	GCCTCTTGGTGGTGTGTTTGATG	63.5	RT-qPCR
*Motilin*-qR	ACTTCTTCCCCGTCTTGCTCTTCT		
*Motilin receptor*-qF	GTTTATCGTTGGCGTCACTGGGA	63.5	RT-qPCR
*Motilin receptor*-qR	GGTCAAAGGGCAGGCAAAGGAAT		
*POMC*-qF	AGCACCACCCTTAGCGTTCT	59	RT-qPCR
*POMC*-qR	ACCTCTTGTCATCCCGCCT		
*NPY*-qF	GCTGGCTACCGTGGCTTTC	59	RT-qPCR
*NPY*-qR	GACTGGACCTCTTCCCATACCT		
*CART*-qF	CGACTGTGGTTGAGAGCCG	55.7	RT-qPCR
*CART*-qR	GACAGTCACACAACTTGCCGAT		
*AgRP*-qF	AGGCTGTGCGTCTCAGTGTC	55.7	RT-qPCR
*AgRP*-qR	GAATCGGAAGTCCTGTATCGG		
*Apelin*-qF	CAGACACGCTGTTTTACACCAC	59	RT-qPCR
*Apelin*-qR	GCACAGCATGGACACCAAGAT		
*NUCB2*-qF	TGGAGACAGACCAGCATTTCAG	55.7	RT-qPCR
*NUCB2*-qR	GGCTCCGTAACCTGTTCACTTC		
*NPFF*-qF	GCGGATGAGCGGGTAATGACT	63.5	RT-qPCR
*NPFF*-qR	GACTCTACCTGCTCCTCGCCG		
*CCK*-qF	GAGGGTAGTCCTGTAGCATCTGA	62.3	RT-qPCR
*CCK*-qR	TTCTACCAGACGAGCCTTTCC		
*Ghrelin*-qF	CCAAGGTGACACGTCGAGATTC	63.5	RT-qPCR
*Ghrelin*-qR	TCCTGATACTGAGATTCTGACATTGAG		
*Gastrin*-qF	GAGTTTCGTCAAGGTATGCGTGT	63.5	RT-qPCR
*Gastrin*-qR	GCAGCCAGTGTCTTCTCCCG		
*EF-1α*-qF	ATGTTCACAATGGCAGCGTC	60	RT-qPCR
*EF-1α*-qR	AAGATTGACCGTCGTTCCG		
*β-actin*-qF	GCCCCACCTGAGCGTAAAT	60	RT-qPCR
*β-actin*-qR	TCCTGCTTGCTGATCCACAT		

## Data Availability

The original contributions presented in the study are included in the article, further inquiries can be directed to the corresponding authors.

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
