# Peer review of "Motilin, a Novel Orexigenic Factor, Involved in Feeding Regulation in Yangtze Sturgeon (Acipenser dabryanus)"

_biomolecules, 2024, doi:10.3390/biom14040433_

Round 1

Reviewer 1 Report

Comments and Suggestions for Authors

Overall this manuscript is well-written, the experiments are well-thought out and the data is convincing. There are a few minor concerns.

 1.      More information about weaning in fish could be added to the abstract to help the reader fully understand how AdMotilin could be a biomarker of adequate nutrition/calories during weaning.

2.      Add cloning to the study goals at the end of the introduction.

3.      In Methods, Section 2.2, more information about weaning, weaning process, diets,etc should be included.

4.      For expression studies, was the whole brain used or where the specific brain regions analyzed? Indicate in the methods and throughout.

5.      For Figure 7, unclear what -6, -3, -1, 0 hours are investigating. Is 0 hours when the fasting begin? If so, what was the rationale for  measuring expression prior to the fast?

6.      Figure 8, the highest dose of AdMotilin increased food intake at 1h, but the middle dose increased food intake at 3h. What is a possible explanation for this finding?

7.      For the IP vs. in vitro administration of AdMotilin, the expression of genes like NPY, AgRP were opposite. Though a potential explanation was discussed in Discussion section, please expand.

8.      Check throughout for typos and missing words.

Comments on the Quality of English Language

Check throughout for typos and missing words.

Author Response

Overall this manuscript is well-written, the experiments are well-thought out and the data is convincing. There are a few minor concerns.

1. More information about weaning in fish could be added to the abstract to help the reader fully understand how AdMotilin could be a biomarker of adequate nutrition/calories during weaning.

Reply: We have added the information in line 20-22.

2. Add cloning to the study goals at the end of the introduction.

Reply: We have added the content in line 121.

3. In Methods, Section 2.2, more information about weaning, weaning process, diets,etc should be included.

Reply: We have added the information in line 149-150.

4. For expression studies, was the whole brain used or where the specific brain regions analyzed? Indicate in the methods and throughout.

Reply: We sampled and analyzed the whole brain, and we have revised the information in the material and methods.

5. For Figure 7, unclear what -6, -3, -1, 0 hours are investigating. Is 0 hours when the fasting begin? If so, what was the rationale for measuring expression prior to the fast?

Reply: The 0 h is the fasing begin time. We measured the expression of AdMotilin and AdMotilinR before feeding time in order to explore their potential role as an orexigenic signal. Several reports have showed that the expression of orexigenic factors such as ghrelin and bdnf were increased before a meal. Reference:

[1] Peddu SC, Breves JP, Kaiya H, Gordon Grau E, Riley LG Jr. Pre- and postprandial effects on ghrelin signaling in the brain and on the GH/IGF-I axis in the Mozambique tilapia (Oreochromis mossambicus). Gen Comp Endocrinol. 2009 May;161(3):412-8. doi: 10.1016/j.ygcen.2009.02.008.

[2] Blanco AM, Bertucci JI, Hatef A, Unniappan S. Feeding and food availability modulate brain-derived neurotrophic factor, an orexigen with metabolic roles in zebrafish. Sci Rep. 2020 Jul 1;10(1):10727. doi: 10.1038/s41598-020-67535-z.

6. Figure 8, the highest dose of AdMotilin increased food intake at 1h, but the middle dose increased food intake at 3h. What is a possible explanation for this finding?

Reply: Thanks for the question. Based on the theory of pharmacodynamics and pharmacokinetics, a higher dose will allow the drug to reach its effective concentration more quickly, reducing the time to onset, so in this result the middle dose increased food intake at 3h, which spent longer time than the highest dose.

7. For the IP vs. in vitro administration of AdMotilin, the expression of genes like NPY, AgRP were opposite. Though a potential explanation was discussed in Discussion section, please expand.

Reply: Thanks for your suggestion, we have added content in the discussion. This difference might be due to the fact that intraperitoneal injection is a peripheral drug delivery method that acts on the brain indirectly, and the appetite factors produced from peripheral tissues may have a negative feedback effect on the expression of NPY and AgRP in the brain. Tissue incubation of Motilin directly affects the expression of NPY and AgRP in different brain regions.The mechanisms of Motilin in regulating feeding may differ between central and peripheral system. The mechanism medicating this kind of interaction in fish is not clarified yet, and further investigation needs to be carried out.

8. Check throughout for typos and missing words.

Reply: We have revised these mistakes in the manuscript.

Reviewer 2 Report

Comments and Suggestions for Authors

Authors need to highlight the importance of this study. The importance to study the molecular mechanisms of feeding of a specific species of fish needs to be better explained. Also, authors probed the mRNA expression of Motilin and its receptor but it would be important to also probe the protein expression and function, so the feeding mechanism involving Motilin would be more throughly understood. 

Author Response

Authors need to highlight the importance of this study. The importance to study the molecular mechanisms of feeding of a specific species of fish needs to be better explained. Also, authors probed the mRNA expression of Motilin and its receptor but it would be important to also probe the protein expression and function, so the feeding mechanism involving Motilin would be more throughly understood.

Reply: Thanks for your kind suggestions. We have added the information about the significance of studying the regulation of feeding by Motilin on the Yangtze sturgeon in the abstract and introduction. This study explored the mRNA expressions of motilin in the weaning and starvation status which suggest motilin might act as a hunger signal in the brain when food is palatable or limited. Furthermore, to explore the possible role of motilin in feeding regulation, this study conducted the in vivo i.p. injection experiment and in vitro incubation test using the mature peptide of Motilin, and found Motilin can increase food intake of Yangtze sturgeon. These results suggest that Motilin peptide has potential application value in improving the decrease of food intake caused by weaning in the seeding cultivation which limits the restoration of wild Yangtze sturgeon population.

Round 2

Reviewer 2 Report

Comments and Suggestions for Authors

Authors answered my questions accordingly.